# Come out of Your Shell—A Comparative Pilot Study for Teaching the Central Plastrotomy in Chelonians Using a 3D-Printed Simulator and a Virtual 3D Simulation

**DOI:** 10.3390/ani15060824

**Published:** 2025-03-13

**Authors:** Marie-Therese Knoll, Andrea Tipold, Michael Pees, Sandra Wissing, Johannes Hetterich

**Affiliations:** 1Clinical Skills Lab, University of Veterinary Medicine Hannover Foundation, 30173 Hannover, Germany; sandra.wissing@tiho-hannover.de; 2Department of Small Animal Medicine and Surgery, Neurology, University of Veterinary Medicine Hannover Foundation, 30559 Hannover, Germany; andrea.tipold@tiho-hannover.de; 3Department of Small Mammal, Reptile and Avian Medicine and Surgery, University of Veterinary Medicine Hannover Foundation, 30559 Hannover, Germany; michael.pees@tiho-hannover.de (M.P.); johannes.hetterich@tiho-hannover.de (J.H.)

**Keywords:** veterinary education, 3D printing, 3D simulation, clinical skills, central plastrotomy, transplastron coeliotomy, chelonians, reptiles

## Abstract

A plastrotomy is a complex surgery technique applied to chelonians (turtles and tortoises). Training in practical skills is critical to master this surgery. Thus, a three-dimensional (3D)-printed simulator and a virtual 3D simulation were designed. Both were tested for their suitability as learning resources in three settings. An objective examination and subjective self-assessment of the student’s skills were conducted alongside an evaluation process of the respective resources. Moreover, veterinary practitioners evaluated the 3D-printed simulator. Both resources were assessed as suitable training devices for a plastrotomy in chelonians. The 3D-printed simulator scored better in all three settings; however, the majority of these findings were not significant. The pilot study indicates that a 3D-printed chelonian simulator is a novel approach with promising potential for practicing surgical access to the coelomic cavity in chelonians.

## 1. Introduction

Reptiles are often referred to as exotic animals [1], and their class is characterized by a remarkable diversity, including over 10,000 species [2]. Recent international studies depict a trend of increasing numbers of reptiles kept as pet animals [3,4], specifically chelonians (turtles and tortoises) [5,6,7,8]. Chelonians represent an extraordinary class of reptiles due to their unique exoskeleton. This anatomical feature connects the dorsal carapace by lateral bone bridges to the ventral plastron [9]. A surgical technique that includes cutting through the plastron and incising the underlying soft tissue can be performed to assess the coelomic cavity. This complex surgery is summed up under the term central plastrotomy or transplastron coeliotomy [10].

Hence, there is an urgent need for a sufficient number of qualified veterinarians to ensure high-quality treatment standards, and veterinary education and research have to be adapted [11]. Although great progress has been evident in various aspects of reptile medicine since the early 1990s [12], there remain challenges regarding exotic animal medicine for both animal owners and veterinarians [13]. Several studies evaluated the knowledge of veterinarians, veterinary students, and animal owners concerning exotic pet welfare in different countries and revealed that changes in the veterinary curriculum, including more teaching about exotic animals, are desirable [11,14,15,16].

The curriculum is impacted by standards defined by the European Association of Establishments for Veterinary Education (EAEVE) [17] or the American Veterinary Medical Association (AVMA) [18]. In order to meet these standards, it is essential that veterinary students are provided with specific practical and soft skills prior to graduation. These so-called “Day One Competences” [17] can be mastered, for example, in clinical skills laboratories [19,20].

In Germany, four of five veterinary clinical skills labs offer reptile-based courses, including hands-on training with snake, lizard, or chelonian simulators for basic procedures like injections, intubation, or blood sample collection [21,22,23,24,25], or online material [23,26,27]. However, the number of courses is still limited compared to the number of courses dealing with other companion animals. This underrepresentation might be associated with the small number of functional commercially available reptile simulators or simulations. A web search comprising nine companies revealed one functional simulator and one simulation for practicing veterinary skills in reptiles apart from anatomical models and taxidermied animals [28,29,30,31,32,33,34,35,36,37].

In this study, a chelonian simulator and a simulation were designed to practice the required steps of a central plastrotomy before transferring these skills to cadavers and live animals. The establishment of both resources describes a novel approach. The haptic chelonian simulator is created by three-dimensional (3D) printing, while the virtual 3D simulation of the chelonian is presented using computer software. Three-dimensional printing is already successfully used in the context of reptile medicine (for instance, in creating 3D-printed prosthetic devices for chelonians) [38,39] or in reptile fundamental research [40,41,42,43]. Various reptile 3D simulations are also implemented in basic research [44,45,46,47], and some are even used for educational purposes, like the interactive anatomical atlas of a chelonian’s head [47].

The current study focuses on three methods to enhance knowledge about the suitability of simulations to teach surgery techniques in chelonians: the evaluation of both resources, an objective structured clinical examination (OSCE), and the self-assessment of skills [48]. This study contrasts each student’s subjective assessment before and after training with their objective OSCE scores. Thus, the aim of our study is the establishment and evaluation of a simulator (S1) in comparison with a simulation (S2), and both are developed for surgical training in chelonians.

## 2. Materials and Methods

### 2.1. Data Privacy

This study was conducted based on the data privacy statement (Datenschutzbestimmung) Art. 6 I 1 lit. e i.V.m. 89 DSGVO, § 3 I 1 Nr. 1 NHG, § 13 NDSG and was reviewed and approved by the Data Protection Officer of the University of Veterinary Medicine Hannover, Foundation (TiHo), Hannover, Germany before the start of this study. An ethical evaluation and approval were granted by the Commission for Research Ethics at the TiHo under the number 25-PK-01.

### 2.2. Developed Resources

A 3D-printed simulator and a virtual 3D simulation were developed for learning the transplastron coeliotomy technique in chelonians in the Clinical Skills Lab (CSL) of the TiHo.

#### 2.2.1. The 3D-Printed Simulator

The 3D-printed simulator (S1) weighed 460 g and consisted of four parts. An exoskeleton of a healthy Tunisian spur-thighed tortoise (*Testudo graeca nabeulensis*) was used as a blueprint for the pantex of the chelonian simulator. The pantex comprises the carapace and the plastron. Both parts were scanned separately with a 3D scanner (SHINING 3D^®^, EinScan Pro2X 2020, Shining 3D Tech Co., Ltd., Hangzhou, China) and printed using a Fused Deposition Modeling (FDM) printer (Ultimaker S5, Ultimaker B.V., Utrecht, The Netherlands). Prior to the scanning process, a calibration was conducted with a pixel error rate of 0.044036. Each item was placed on a turntable (SHINING 3D^®^, Industrial Pack, Shining 3D Tech Co., Ltd., Hangzhou, China), while the scanner was fixed on a tripod (SHINING 3D^®^, Industrial Pack, Shining 3D Tech Co., Ltd., Hangzhou, China). Forty scans were taken of each object (twenty dorsally and twenty ventrally). Afterwards, the generated data were exported to a slicing software (UltiMaker Cura version 5.7.2, Ultimaker B.V., Utrecht, The Netherlands). The following settings were selected: heating temperature: 215 °C, printing speed: 70 m/s, infill density: 80%, and resolution: 0.1 mm. During the process, the actual life size of the chelonian was preserved continuously. The printing was conducted by dual extrusion using a main material (Ultimaker Tough PLA black 2.85 mm 750 g, Ultimaker B.V., Utrecht, The Netherlands) and a support material (Ultimaker Breakaway support material white 2.85 mm 750 g, Ultimaker B.V., Utrecht, The Netherlands). The supporting structures were removed with pliers.

The carapace was printed within 36 h and 51 min. The consumption of material consisted of 156 g/19.98 m in total. Modifications were made to improve the appearance of the carapace. An airbrush gun (SPARMAX, DH-3 nozzle size: 0.3 mm, Harder & Steenbeck, Norderstedt, Germany) attached to an airbrush compressor (SPARMAX, AC-55, Harder & Steenbeck GmbH & Co. KG, Norderstedt, Germany) was used to spray the shell. In addition, the carapace was sprayed in light brown (model color, acrylic colors, 70.819, Iraqi sand, 17 mL, ACRYLICOS VALLEJO, S.L., Barcelona, Spain). Two further colors (model color, acrylic colors, 70.941, burnt number, 17 mL, ACRYLICOS VALLEJO, S.L., Barcelona, Spain; model color, acrylic colors, 70.994, dark gray, 17 mL, ACRYLICOS VALLEJO, S.L., Barcelona, Spain) were manually painted onto the carapace. A treatment with clear spray (EDDING^®^, 5200 permanent spray clear lacquer, glossy, edding International GmbH, Ahrensburg, Germany) fixated the colors.

The plastron was printed with the same settings and materials as described above. Modifications were performed for improved stability of the pantex. The elongation of the lateral sidewalls was achieved by pieces of metal (6 cm × 0.5 cm × 5 cm length × width × height) and monster clay (Monster Clay, Premium Grade, SOFT/1, The Monster Makers, Inc., Cleveland, OH, USA). Scanning and printing of the modified plastron proceeded as previously described. The final printing time of the modified plastron comprised 16 h and 23 min. The consumption of material was 156 g/19.98 m in total. Four holes were drilled through the carapace and the elongated sidewalls of the plastron. They had a distance of 4 cm from each other and 2 cm from the bottom of the plastron.

A replica of the soft tissue was created from translucent silicon (Ecoflex, 00-50, KAUPO, Plankenhorn e.K., Spaichingen, Germany) infused with brown pigments (Silc-pig, brown, pantone, 4625C KAUPO, Plankenhorn e.K., Spaichingen, Germany). The measurements (head–tail) were 25 cm × 5 cm (length × width), including a deepening that resembled the coelomic cavity. The deepening comprised 12 cm × 4 cm (length × width). Five organ models were designed from colored balloons (EVERTS, Décor Line, 13 cm, Round, Standard Yellow Sunshine; EVERTS, Décor Line, 115 cm Modelling E360, Fashion Ocean Blue; EVERTS, Décor Line, 115 cm Modelling E260, Standard Festive Green; GLOBOS, water ballons, Globos-Luftballon GmbH, Hamburg, Germany). The balloons were filled with fine sand (Best for Garden, 25 kg, sandpit sand, quartz sand, grain size 0–1 mm, 4mybaby GmbH, Görlitz, Germany) and secured with super glue (PATTEX^®^, instant adhesive, Henkel AG & Co. KGaA, Düsseldorf, Germany). Color coding enhanced their differentiation (heart = red, liver = yellow, urinary bladder = orange, small intestine = blue, and colon = green). These organ models were placed inside the deepening of the soft tissue replica.

The organ models were covered with a tube of latex tissue mimicking the coelomic membrane. Therefore, the latex tissue (RadicalRubber, latex-sheeting per meter, transparent, 0.20 mm, London, UK) was cut into pieces (9.5 cm × 20 cm). Each piece was glued together with a latex adhesive (adhesive for latex thinner than 0.6 mm) to create a tube. Each tube was painted with a blue marker (EDDING^®^, 400 Permanent marker, 1 mm tip, color 003 blue, edding International GmbH, Ahrensburg, Germany) to display the two abdominal veins. Each vein was approximately 0.5 cm thick. The simulated veins were spaced 2 cm apart and 2 cm from the edges of the latex tissue piece. Finally, all parts were assembled (Figure 1). The soft tissue replica, including the organ models and the coelomic membrane, was placed inside the plastron. The carapace was secured to the plastron with two screws (SPAX^®^ Z2, 3.5 × 16 International, Ennepetal, Germany) on each side.

#### 2.2.2. Virtual 3D Simulation

The virtual 3D simulation (S2) was designed using Materialise Mimics Innovation Suite 26.0.0.576 (Materialise NV, Leuven, Belgium) (Figure 2). Computed tomography (Philips IQon Spectral CT, Philips Healthcare, Philips GmbH, Hamburg, Germany) imaging of a healthy physiological chelonian was obtained from the Department of Small Mammal, Reptile and Avian Medicine and Surgery at the TiHo. The data were imported into the software program, and anatomical structures were separated automatically according to their density. As both plastron and carapace had the same density, they were separated manually using the segmentation tool. The first segmentation separated the pantex from the remaining bones. The pantex was separated again into the carapace and the plastron following the suturae (bone lines). The minimum of Hounsfield units (HUs) was, therefore, set to 160 HUs, and the maximum was set to 3070 HUs. A mask (color) was selected for each segment, and annotations were edited. Both annotations and masks could be hidden. The 3D simulation of the chelonian was presented in a four-field view including the axial, sagittal, and coronal plane. The visual presentation of the chelonian was selected within the modus “volume rendering”. The visualization included the options “Soft Tissue”, “Bone & Skin”, “Bone”, or “Bone Transparent”. The virtual chelonian could be moved and rotated to mimic the correct positioning of the animal during a plastrotomy. Moreover, the tool “Distance over the surface OP“ in the menu “Measure” was selected to mark a trapezoid cut out on the plastron for the surgery.

### 2.3. Study Design

#### 2.3.1. Study with Students from Semester 5 to 8

A total of 29 TiHo students on the 5th to 8th semester participated in this study from the summer semester of 2023 until the summer semester of 2024. This study was integrated into the elective class “reptile surgery”, where students performed various surgeries on reptile cadavers, excluding a plastrotomy under the supervision of a veterinarian specialized in reptile medicine (JH) from the Department of Small Mammal, Reptile and Avian Medicine and Surgery at the TiHo. In each of the three semesters, ten veterinary students were selected and randomly assigned to two groups consisting of five people each (Figure 3). Students in Group 1 were assessed in semester 1 (S1), while students in Group 2 were assessed in semester 2 (S2). All students participated in a two-hour preparation course in the CSL at the TiHo. A theoretical introduction to a central plastrotomy was presented during the first 60 min of the course. The remaining 60 min dealt with the hands-on training of a central plastrotomy using the respective resources. Group one used the material and tools to prepare S1 for the central plastrotomy by drawing incision lines on the plastron. The plastron was cut open using a dremel (Dremel 3000, DREMEL^®^ EUROPE, BOSCH POWER TOOLS B.V., Breda, The Netherlands) and reclosed by epoxy resin and fiberglass fabric (both: NIGRIN, MTS GROUP, INTER-UNION TECHNOHANDEL GMBH, Rülzheim, Germany). Group two highlighted the anatomical structures as depicted in Figure 2, positioned S2 for the conduction of a transplastron coeliotomy, and drew incision lines on the virtual plastron.

Prior to the course, all students completed a questionnaire concerning their demographic data, previous knowledge, and experience (DKE) (see Appendix A). Additionally, all students had previously completed a self-efficacy (SE) questionnaire prior to the course to record their self-assessment of skills (see Appendix A). It was handed to the students at three temporal stages (t1 = prior to the preparation course, t2 = after attending the preparatory course, t3 = after performing the OSCE). Furthermore, the students completed a questionnaire concerning the evaluation of the respective resource (ES1) after the course (see Appendix A). All students demonstrated their acquired practical skills independently in an OSCE setting seven days after the initial preparation course. The task focused on the performance of a central plastrotomy on a chelonian cadaver providing access to the coelomic cavity. It was neither possible to provide cadavers from only one chelonian species nor to select the cadavers by sex or size due to the limited number of cadavers available. The cadavers were numbered and allocated by random selection using a randomizer. The shell thickness of the abdominal shield of the plastron was measured post-surgery. Further surgical soft tissue procedures, like biopsy sampling or ovariohysterectomy, were not included, as the focus was the application of the transplastron surgery technique as described in the literature [37]. The assessment of skills was based on a checklist (CL) (see Appendix A). The examiner was an independent veterinary practitioner with experience in performing plastrotomies in reptiles and conducting an OSCE. The examiner neither took part in the preparation course nor was aware of the students’ group affiliation.

#### 2.3.2. Evaluation by Experienced Veterinarians

The 3D-printed chelonian simulator was tested and evaluated by 10 veterinarians of the Department of Small Mammal, Reptile and Avian Medicine and Surgery at the TiHo. These experts independently performed the surgical technique on the model. A separate questionnaire was used to record the evaluation by experienced veterinarians (ES2) (see Appendix A).

### 2.4. Data Acquisition

#### 2.4.1. Questionnaires

The questionnaires used in this study contained a privacy statement and an individual code (chiffre). The coding was necessary to assign questionnaires to the same person without violating the privacy statement. Different types of questions were used in the questionnaires, including checkbox questions with single-choice and multiple-choice answers alongside free-text responses. Likert-type questions with a four-point scale (1 = “I strongly agree”, 2 = “I agree”, 3 = “I disagree”, 4 = “I strongly disagree”, and 0 = “no indication”) were predominantly used. School grades (1 = very good, 2 = good, 3 = satisfactory, 4 = sufficient, 5 = poor, and 0 = ”no indication”) were used once to assess the fidelity of the respective resource. A questionnaire about the students’ demographic data and their previous knowledge and experience was compiled (DKE). The DKE comprised a total of 20 items. Furthermore, a questionnaire about the students’ self-efficacy (self-assessment of skills) (SE) was compiled, comprising 15 items. Moreover, two questionnaires for the evaluation of the respective resources were compiled. The questionnaire for the student groups (ES1) consisted of 33 items. The questionnaire for the clinicians (ES2) (see Appendix A) comprised thirty-four items, including one additional item concerning their professional designation. Variants of these questionnaires for data collection had already been validated for previous study purposes in the CSL [49,50]. Prior to their use, a quality check including comprehension, form, didactics, and time taken to complete the questionnaires was conducted by the E-Learning Service of the TiHo. The Data Protection Officer of the TiHo approved all documents used for the evaluations.

#### 2.4.2. Objective Structured Clinical Examination (OSCE)

An OSCE was conducted following a procedure already validated in previous studies conducted by the CSL [49,50]. The learned clinical skills of both student groups were examined within a 20 min setup. The students completed a task, which was presented in the student sheet (S), including the selection of a radiographic image. Two radiographic images were obtained from the Department of Small Mammal, Reptile and Avian Medicine and Surgery of the TiHo. Picture A depicted a chelonian with pathological dystocia; picture B depicted a healthy, physiological chelonian. The students were asked to select the correct picture presenting an indication of a plastrotomy. A plastrotomy was performed by the students on a chelonian cadaver following the procedure of the preparation course. An independently trained member of the teaching staff conducted the examination using CL. The checklist comprised an individual code (chiffre) and 32 items. Thus, a step-by-step performance of the surgical technique in chronological order was guaranteed. The CL was divided into pre-surgery, surgery, and post-surgery procedures. In addition, items concerning the handling of the instruments and an antiseptic working practice were part of the CL. Practical skills performed completely and correctly were considered “fulfilled”. Practical skills not performed within the time limit were considered “not fulfilled”. The additional option “partly fulfilled” was available for five items. An example was the preparation of material, which was marked as “partly fulfilled” when the student failed to prepare one item before moving on to the next step. However, the task was noted as “not fulfilled” if the student failed to prepare two or more of the above items. Variants of such checklists had already been validated for previous studies conducted by the CSL [49,50]. Prior to use, a quality check including comprehension, form, and didactics was conducted by the E-Learning Service of the TiHo. Additionally, the CL was reviewed with a focus on scoring by four veterinarians from the CSL of the TiHo. Moreover, an audit was performed by two clinicians from the Department of Small Mammal, Reptile and Avian Medicine and Surgery of the TiHo, with a focus on the professional accuracy of the CL. The Data Protection Officer of the TiHo approved the use of the CL.

### 2.5. Statistical Analyses

Data analysis was performed using SAS software, version 9.4, and SAS^®^ Enterprise Guide^®^ 7.15 (SAS Institute Inc., Cary, NC, USA). A descriptive analysis and graphic creation were conducted with Microsoft^®^ Office Excel 2016 (Microsoft Corporation, Redmond, WA, USA). The *p*-values below 0.05 were assumed to be significant.

#### 2.5.1. Study with Students from Semester 5 to 8

A Shapiro–Wilk test on a normal distribution was conducted, concluding that the data were not normally distributed. The OSCE results were analyzed with descriptive statistics. The Wilcoxon signed-rank test for two non-linked random samples was selected to compare the OSCE results of the two groups with each other. Testing for Spearman’s rank-order correlation was conducted regarding the causality between the shell thickness and the OSCE results. The self-assessment of the student’s skills was analyzed, including standard deviation, the mean, the minimum and maximum values, and the median at three temporal stages in this study. Additionally, a distribution analysis for signed rank was performed with the two groups at all points in time. The subjective self-assessment of the students was compared with the objective assessment by the examiner in the OSCE setting. Testing for Spearman’s rank-order correlation was applied to compare the OSCE scores with the self-assessment sum of points after the preparatory course. In this way, it could be determined whether the students who were generally more confident in their practical skills were also able to demonstrate more skills correctly in the examination.

#### 2.5.2. Evaluation Analysis

A descriptive analysis was conducted, and the results of the three evaluation results were depicted graphically. The fidelity was presented in bar charts, while Likert-type items were represented with diverging bar charts. The semester affiliation by group was depicted in a bar chart (see Appendix A). Furthermore, descriptive analysis was applied to the DKE.

## 3. Results

### 3.1. Study with Students in Semester 5 to 8

In this study, 29 questionnaires (DKE, SEt1, and SEt2) were evaluated. A total of 28 checklists and questionnaires (SEt3) were interpreted because one person took part in the preparatory course but did not attend the OSCE. Appendix A depicts the semester affiliation for students in both groups at the time of the elective. Further findings of DKE are depicted in Table 1. Four persons in Group 1 (26%) and two persons in Group 2 (14%) had already completed professional training as veterinary assistants prior to their study. Contact with reptiles in private life (for instance, as pets) was only confirmed by one person in Group 1 (6%) and by two persons in Group 2 (14%). While chelonians were selected twice within Group 1, both snakes and lizards were selected once within Group 2. None of the students attended the elective course “reptile surgery” before taking part in this study. All 28 students, regardless of their group affiliation, experienced theoretical contact with reptiles during their veterinary studies and welcomed further simulation/simulator-based training on reptiles and surgery. Hands-on contact with reptiles during the veterinary studies was experienced by 12 persons (80%) in Group 2 and 13 persons in Group 1 (92%). With the exception of one student, participants had no practical experience in performing a central plastrotomy. It can be concluded that the experience level of the participants in this study varied slightly.

#### 3.1.1. Objective Structured Clinical Examination (OSCE)

A total of 28 students took part in the OSCE. Fifteen people were assigned to Group 1 assessing S1. Thirteen students were allocated to Group 2 assessing S2. A maximum score of 63 points could be achieved in the OSCE. The lowest score was 25% (16 points), achieved once in Group 2. The highest score was 98% (62 points), achieved twice in Group 1. The standard deviation was 21.7. In Group 1, a minimum of 45% (28.5 points) and a maximum of 98% (62 points) were scored. The median was 83% (52.0 points) and the mean score was 15.7 (Table 2).

In Group 2, a minimum of 25% (16 points) and a maximum of 92% (58 points) were scored. The median was 73% (46.5 points) and the mean score was 13. Three of the twenty-eight students failed the OSCE, achieving less than 60%. All three students belonged to Group 2. A total of 14 students did not finish the task within the set timeframe (seven in Group 1 and seven in Group 2). Despite the mean difference of 6.6 percentage points in the OSCE, the performance of students who completed the training using S1 was not significantly different from the students trained with S2 (Table 2).

#### 3.1.2. Correlation of Shell Thickness and OSCE Results

The shell thickness of the abdominal shield of the plastron was measured to estimate its influence on the OSCE results. Ten measurements of the shell thickness were collected in Group 1 and nine in Group 2. In both groups, the shell thickness varied between a minimum of 0.3 cm and a maximum of 0.6 cm with a median of 0.4 cm. The shell thickness and the OSCE results in both groups showed a negative correlation. A thinner shell contributed to a higher OSCE result. Despite the negative correlation, these findings were not significant (Table 3).

#### 3.1.3. Self-Assessment of Skills

Self-efficacy was assessed at three different points in time: before the preparatory course (t1), before the OSCE (t2), and after the OSCE (t3). A total of 29 students took part in this self-assessment of skills at t1 and t2. Fifteen students belonged to Group 1 and fourteen students belonged to Group 2. At t3, the number of participating students in Group 2 decreased to 13. It must be noted that higher values corresponded with a lower self-assessment of skills. Significant differences in the SE mean score and the sum of scores are depicted in Table 4.

The mean values in Group 2 were higher than the mean values in Group 1 for 14 of the 15 items at t1. Only item 6 dealing with the preparation of surgical instruments and materials revealed a lower mean value in Group 2 at t1 (Appendix A). The difference between the SE of the groups was not significant at t1 (*p*-values all above 0.05).

In addition, the mean values of Group 2 were at t2 higher for all 15 items than the mean values in Group 1. A significant difference between the SE of the groups was only given for item 7 at t2 (Appendix A). Students trained on S1 were significantly more confident in correctly estimating the depth of the cutting disk used in a plastrotomy compared to students trained on S2 (*p* = 0.0362).

The mean values at t3 were lower in Group 1 than in Group 2 for 14 of the 15 items. The only exception was item 4 concerning the determination of a medical indication for performing a plastrotomy based on a radiographic examination (Appendix A). That difference was not significant. Significant differences regarding the SE between the groups at t3 were detected for item 12 and item 13. Thus, students trained on S1 had significantly more confidence in performing a plastrotomy on a chelonian cadaver with (*p* = 0.0250) and even without (*p* = 0.0313) veterinary supervision and guidance than students trained on S2.

Additionally, there was a significant difference between the self-assessment of skills from t1 to t3 in 14 of the 15 items for both groups. The mean values decreased significantly from t1 to t3 in seven of the fifteen items for both groups. Contrarily, the mean values for the self-assessment of skills increased significantly in Group 1 regarding item 2 (knowing the steps for preparation, conducting, and post-processing of a plastrotomy) and item 8 (selecting the right angle for the cutting disk) from t1 to t3.

In Group 2, the mean values for the self-assessment of skills increased significantly for item 5 and item 6 (knowing what instruments and materials to prepare for a plastrotomy) and item 11 (confidence to seal the plastron after a plastrotomy) from t1 to t3. A non-significant (*p* = 0.0781) increase in mean values for the self-assessment of skills from t1 to t3 concerning item 12 (confidence to conduct a plastrotomy on a chelonian cadaver with guidance of a veterinarian) was identified for Group 2. Furthermore, there was a significant increase in the mean values for the self-assessment of skills for item 9 (naming relevant anatomical structures within the coelomic cavity correctly) and item 10 (knowing what anatomical structures can be damaged in a plastrotomy conducted wrongly) in both groups from t1 to t3.

It can be concluded that students trained on S1 assessed their skills on average to be higher at all three temporal stages and thus were more confident than students trained on S2.

#### 3.1.4. Correlation of Self-Assessment of Skills and OSCE Results

A total number of 28 students took part in the OSCE, and 29 students assessed their self-efficacy after the preparatory course and before the OSCE (Table 5). The higher the sum of points in the SE, the lower the student’s self-assessment of skills. That means students in Group 1 with an overall lower sum of points had a higher self-assessment of their skills overall after the preparation course. A positive correlation between the sum of points at t2 and the OSCE score for students in Group 1 and a negative correlation between the sum of points at t2 and the OSCE score for students in Group 2 was identified. In addition to these correlations, the findings were not significant for both groups, with *p*-values above 0.05.

Finally, it can be concluded that neither group affiliation, student’s self-assessment of skills nor shell thickness alone, led to a significant difference in the OSCE results.

### 3.2. Evaluation

In this study, 28 questionnaires of students (ES1) were collected. A total of 15 students evaluated S1, while 13 students assessed S2. Additionally, 10 questionnaires of experts (ES2) were interpreted. All students in Group 1 (100%, *n* = 15) rated their enjoyment when learning how to perform a central plastrotomy in chelonians using simulators as “good”. However, the majority of students in Group 2 (53%, *n* = 7) rated this aspect only as “mediocre”. All students in Group 1 (100%, *n* = 15), as well as most students in Group 2 (93%, *n* = 12), stated that it was useful to train the procedure of a plastrotomy on a simulator before performing it on a chelonian cadaver. Moreover, all students in Group 1 (100%, *n* = 15), as well as most students in Group 2 (84%, *n* = 11), assumed it would be useful to train the procedure of a plastrotomy on a simulator before performing it on a living chelonian. S1 was, according to the majority of students, a suitable resource for both veterinary students and veterinarians to train for a central plastrotomy. In contrast, only one-third of students in Group 2 agreed with that statement regarding S2. S1 enhanced the learning success, self-confidence, and safety of its users according to students in Group 1.

Most students did not understand the respective resource (S1 or S2) as a substitute for the use of cadavers. However, most students in Group 1 evaluated S1 as a useful addition to already existing resources. Concerning S2, only half of the students in Group 2 agreed on that point. Also, S2 enhanced the learning success, self-confidence, and safety of its users. However, only half of the students in Group 2 agreed with that statement. When asked which resource the clinical experts favored, the majority (60%, n = 6) chose S1. The majority of both student groups (Group 1: 67%, *n* = 10, Group 2: 85%, *n* = 11) preferred a combination of S1 and S2. In contrast, S1 alone ranked first by the experts and second by the students, while S2 alone was not selected by anybody in the three groups (Table 6). Further research concerning combined training with both S1 and S2 to practice this surgery technique is needed.

#### 3.2.1. Evaluation of the 3D-Printed Simulator (S1) by Students in Group 1

The fidelity of S1 was rated using school grades and is depicted in Figure 4. The majority of students rated the following six aspects as “very good”: functionality, user-friendliness, visibility of landmarks, optics, size, and haptics. Three aspects were rated as “sufficient”. The least positive rating was identified for reusability, as it was mostly rated as “satisfactory” or “poor”.

Furthermore, the evaluation of free-text responses revealed that the students criticized the material chosen for the 3D-printed plastron since it was prone to melting while using the dremel.

A total of 13 Likert-type items concerning the application and effects of S1 were evaluated. The majority of students in Group 1 agreed or strongly agreed with seven of the eleven items. Most of the students disagreed or strongly disagreed with the statement that S1 functioned as a useful substitute for chelonian cadavers when practicing the central plastrotomy. The option “strongly disagree” was chosen concerning two items by the students. A selection of the evaluated Likert-type items is shown in Figure 5.

#### 3.2.2. Evaluation of the Virtual 3D Simulation (S2) by Students in Group 2

The fidelity of S2 was rated using school grades. The results are shown in Figure 6. The majority of students rated three aspects, namely, the reusability, the haptics, and the visibility of landmarks as “very good”, while only half of the students considered the anatomical correctness to be “very good”. Five aspects achieved the rating “sufficient”. The user friendliness was evaluated mostly as “satisfactory”, thus receiving the least positive rating.

The evaluation of Likert-type items on the application and effects of S2 comprised 13 aspects. Most of the students in Group 2 agreed or strongly agreed with only two of the eleven items. Furthermore, the option “strongly disagree” was selected four times. It can be noticed that the answers given by students in Group 2 were not as homogeneous as the answers given by students in Group 1. The majority of students in Group 2 nevertheless disagreed or strongly disagreed with the statement that S1 functioned as a useful substitute for chelonian cadavers when practicing the central plastrotomy, just like students in Group 1. A selection of the evaluated Likert-type items is depicted in Figure 7.

#### 3.2.3. Evaluation of the 3D-Printed Simulator (S1) by Experts

The fidelity of S1 was rated using school grades. All 11 items regarding the fidelity of S1 were rated mostly with “very good” or “good”. Hence, the experts validated S1 even better than the students in Group 1. The visibility of landmarks, however, was the only aspect to be rated “poor”. Moreover, a critique of the material condition of S1 was identified in the free-text responses.

## 4. Discussion

The establishment, evaluation, and testing of a 3D-printed chelonian simulator compared to a virtual 3D simulation is a novel approach. To the authors’ knowledge, no comparable simulator for this purpose has yet been described in the literature. Novel learning resources are on the rise, yet the assessment of their suitability in general and the comparison of the results after implementing the resources in specific is scarce [51,52]. In the current study, it was shown that S1 and S2 are both suitable for practicing the central plastrotomy in chelonians. The simulator, however, performed on average better in all test environments than the simulation. Students trained with S1 scored a higher OSCE result than students in Group 2. None of the students in Group 1 failed the examination, i.e., achieving less than 60% of the points. Moreover, the students using S1 for preparation had an overall higher confidence in their skills. A significant difference in the SE was detected between the groups for item 7 at t2 as well as item 12 and item 13 at t3. The comparative evaluation of the resources revealed that S1 received positive feedback and high-fidelity ratings from both veterinarians and veterinary students.

Nonetheless, this study also demonstrated that there was no significant difference in OSCE performance regarding the self-assessment of skills of most aspects between the two groups. As neither the group affiliation, the shell thickness, nor the SE alone is able to fully explain the OSCE results, there must be other factors elucidating the performance of students conducting a central plastrotomy on a chelonian cadaver. The handling of the instruments and the individual disposition of each student may serve as an important factor. Unfortunately, the manual dexterity of the students was not assessed in the DKE. Thus, further research regarding the influence of craftsmanship on the OSCE results is needed.

### 4.1. Comparison of Novel with Traditional Educational Resources

The fact that the mean values for the students’ self-assessment of skills increased significantly from t1 to t3 regarding seven aspects might be explained by the differences between a simulator/simulation and a cadaver. Replicas might never be able to fully mimic a cadaver, including sensual experiences, like smell and haptic characteristics. A false understanding of cadaveric materials and their handling could be the consequence when only trained with 3D-printed models [53]. The self-assessment of skills after performing the surgery technique on a chelonian cadaver might be the most realistic, as it is not limited to a visual (S2) or haptic (S1) experience. Students reconsidered their subjective estimation of skills after the OSCE, thus resulting in a more accurate self-assessment.

These findings are in line with the evaluation of this study where students and experts denied that the respective resources can substitute the use of cadavers to practice a central plastrotomy. A survey conducted amongst veterinary students supports these findings where cadavers were favored to learn the anatomy [51]. The dissection of cadavers is also considered to be the “gold standard” for teaching anatomy [52]. Hence, the use of animal carcasses is still of great importance for educating veterinary students.

Nevertheless, handling cadavers causes more anxiety than 3D-printed alternatives [54]. Legal regulations must be taken into consideration for buying, storing, and handling the cadavers [54]. Also, ethical, cultural, or religious concerns regarding the educational use of carcasses are minimized using other resources [54]. Both S1 and S2 offer advantages over traditional teaching devices, including improved hygiene. While S1 can be used in every room with sufficient ventilation, S2 can be used on any device with access to the software program. This also includes the potential to practice the central plastrotomy at home. In contrast, cadavers must be cooled and used in separate rooms, like section halls, to avoid contamination [55].

Additionally, there is a greater availability of both resources, as the number of carcasses is often limited [55]. That aspect was also present in this study where it was not possible to select the chelonian cadavers by sex, health status, age, or species for the OSCE.

By German law, it is allowed to conduct animal testing, which is undertaken for training, further education, or training purposes [56]. This aspect also comprises the education of veterinary students at the TiHo. However, it is also required by the same law to limit animal experiments to what is absolutely necessary, including the obligation to improve the methods used in animal experiments [56]. The possibility to repeat, improve, and master the clinical skills needed for this surgical intervention on a simulator/simulation contributes to animal welfare. Species conservation aspects are also positively influenced using novel resources since fewer animal cadavers are needed for practicing the transplastron technique. The concept of 3Rs (Replacement, Reduction, Refinement), as mentioned in 2.2. in the “Day One Competences” [17], is applied when using alternate teaching devices, like simulators and simulations.

Another aspect is the anatomical correctness of both resources. CT scans from a chelonian were used to design S2. The exoskeleton of a healthy Tunisian spur-thighed tortoise (*Testudo graeca nabeulensis*) was scanned for S1. Especially, the analogy to reality was highlighted by both students and experts in the evaluation. However, it is also possible to use other CT data for S2 or to modify the data used for S1. Thus, the size or printing material can be adjusted to fit the circumstances when practicing for an upcoming surgery [55].

Moreover, both resources are considered to be sustainable. S2 can be used without causing any waste. The carapace and the organ models of S1 can be reused after performing a central plastrotomy, while the plastron and the artificial coelomic membrane need to be replaced. Moreover, the filament used to print the exoskeleton is biodegradable under industrial composting conditions [57].

Furthermore, S1 and S2 offer a standardized teaching device with a constant quality. Thus, it is possible to measure progress after performing the surgery multiple times under the same conditions. In addition, the physiological organ models of S1 can be exchanged with other organ models that mimic, for instance, dystocia. In this way, the performance of different techniques can be compared with each other in an OSCE setting. Hence, it can be concluded that chelonian simulators/simulations function as a useful addition to already existing teaching material.

### 4.2. Correlation of Subjective Self-Assessment and Objective Performance

Regarding the causality between the self-assessment of skills and the OSCE scores, different outcomes for the respective group of students were identified. A negative correlation between the sum of points at t2 and the OSCE scores for students in Group 2 and a positive correlation between the sum of points at t2 and the OSCE scores for Group 1 students were projected. Although both results were not significant, it can be concluded that students in Group 2 were able to assess their skills more accurately than students in Group 1. The subjective estimation of Group 2 students mirrored their objective performance in the examination. Students in Group 1 were confident but tended to overestimate their skills. The overestimation of a person’s own competence is summed up under the Dunning–Kruger effect [58]. While some studies proved that the self-assessment of skills of participating students was higher than their objective performance [49,51], others explored a positive correlation between high self-assessment with high OSCE scores [50].

Bandura established the concept of self-efficacy [48]. The term describes the confidence a person has in their ability to perform a certain skill or task. A high level of self-efficacy positively affects an individual’s mindset concerning their confidence in solving problems independently, thus resulting in a higher success rate [48]. A person with poor self-efficacy is impaired by that low level of confidence and, therefore, not able to show their full potential when performing a skill or task [48]. This thesis can be proven by students in Group 2, where the participants with a higher self-assessment of skills also achieved higher OSCE scores than the ones with a lower self-assessment of skills within that group.

Another finding of this study was the ranking of the resources in terms of the respondents’ desire to use them in preparation for a central plastrotomy. Both the experts and the two student groups rated the resources from best to worst: a combination of S1 and S2, only S1, and only S2. That finding requires further research, for instance, using a cross-over design. However, other studies comparing 3D-printed models with other resources already proved that using 3D-printed simulators as a training device offer the most promising results [51,53]. Students using 3D-printed cardiac replicas achieved significantly higher post-test scores than students using cadaveric material or a combination of both resources [53]. Additionally, the scores of students using a 3D-printed hoof model for preparation were significantly higher compared to the ones from students using textbooks or a 3D simulation. Scores from the textbook and the 3D simulation group did not show a significant difference, indicating that these resources offer the least promising results [51]. That fact might explain why S2 was not selected as a desired tool for practicing a transplastron coeliotomy by both student groups and the experts. Another explanation may be found in the evaluation of the resources. S1 was rated better overall than S2, and the answers given by students in Group 1 were more homogeneous than those of the students in Group 2.

### 4.3. Limitations and Strengths

A sample size calculation was conducted prior to the start of this study. The calculation was based on a testing trial with 10 veterinary students trained on an S1 prototype who performed a central plastrotomy in an OSCE setting. The students’ feedback was collected, and modifications concerning the simulator and the general procedure were put into practice. A total of 15 students per group were assumed to be sufficient to deliver significant results. However, these expectations differed from the actual results in this study. The fact that one person did not take part in the elective course and another student did not participate in the OSCE was another limitation concerning the sample size. Moreover, the total sample size of 30 students was relatively small. During this study, the existing schedule and the COVID-19 (coronavirus disease 2019) regulations limited both the number of persons per group and the electives per semester. Hence, this pilot study predicts trends that need to be confirmed with greater cohort sizes. However, other studies with a focus on the suitability and accuracy of 3D-printed replicas where an evaluation was conducted by an expert group included five [59] to ten persons [60]. Other studies with a similar design comparing 3D-printed models with other resources included relatively small cohort sizes of between 16 and 24 persons per student group and reflected the sample size calculation of the current study [51,53].

The choice of material for S1 was criticized by both experts and students since small gleans are created when cutting through the 3D-printed plastron. That process releases heat, which causes the material to melt instead of breaking like a bone. When the material has cooled down again, it cannot be avoided that these small gleans sometimes reconnect themselves with the plastron and the cuts need to be intensified. However, the material used for 3D printers based on Fused Deposition Modelling (FDM) must be heat sensitive to be printable. The filament is heated in the print core and gains more flexibility. Afterwards, the material is pressed through a nozzle onto the building plate where it cools down and remains in a rigid and solid state [61,62,63]. Such FDM material has been used successfully in the CSL in the past. For instance, 3D-printed teeth are used in a commercial horse head simulator for dental procedures where the instruments also release heat [64]. Additionally, several studies in both human and veterinary education have also concluded that 3D-printed models can serve as useful resources with a focus on surgery or anatomy [51,52,53,54,55,60]. Low material costs and rapid prototyping represent positive properties for choosing such models as veterinary education devices [60]. Finally, the plastron of a live chelonian is always cooled when working with heat-releasing instruments or detergents [10]. Sterile saline solution can be used to prevent damage to the bone tissue during a central plastrotomy [10,65]. In this study, the cooling was not conducted actively since the students were asked to perform the central plastrotomy in the OSCE independently. However, each student had to mention the cooling process in the examination. The fact that heat affects the plastron negatively is an important learning step for the students, which can also be experienced when using S1. Future implementation of the learning resource could include a different setting where working in a team and actively conducting cooling of the plastron is conceivable.

### 4.4. Further Implementation and Perspective of S1

The 3D-printed chelonian simulator will be further implemented at the TiHo in a separate elective course “chelonian plastrotomy”. A tutor will guide small student groups through the procedure of a central plastrotomy using S1. The presentation of the preparation course alongside access to an educational video will be provided on the platform TiHo Moodle. Moreover, S1 will be used in the training of veterinary interns at the Department of Small Mammal, Reptile and Avian Medicine and Surgery at the TiHo to practice independently for upcoming surgery. Finally, the simulator could possibly be modified after the development of the corresponding 3D printers and thus become more lifelike so that the desire to use cadavers is further minimized.

## 5. Conclusions

The use of 3D-printed simulators and virtual 3D simulations in a protected learning environment, like a clinical skills lab, depicts a novel approach to learning complex surgical skills, like a central plastrotomy. The findings in this pilot study provide the groundwork for further research, such as the development of new simulators, reflecting the natural conditions of the animals.

## Figures and Tables

**Figure 1 animals-15-00824-f001:**
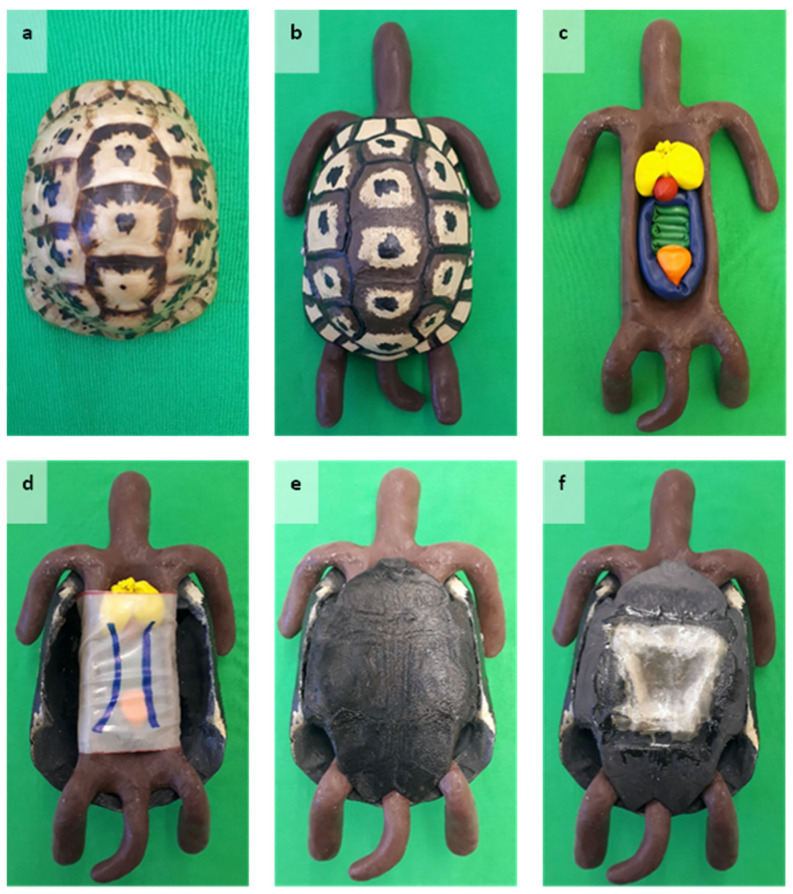
The 3D-printed simulator (S1): (**a**) carapace and plastron of a Tunisian spur-thighed tortoise (*Testudo graeca nabeulensis)*. (**b**) Complete S1 (dorsal view). (**c**) Silicone body with organ models (ventral view). (**d**) Assembling of the carapace, artificial coelomic membrane, silicone body, and organ models (ventral view). (**e**) Complete S1 (ventral view). (**f**) S1 post-plastrotomy (ventral view).

**Figure 2 animals-15-00824-f002:**
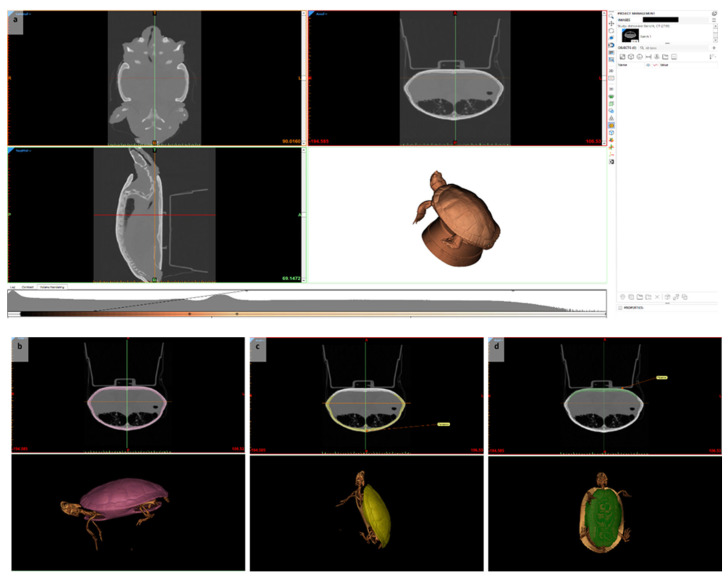
Virtual 3D simulation (S2): (**a**) soft tissue visualization, (**b**) segmented pantex depectied in fuchsia, (**c**) segmented carapace decpicted in gold with annotation, (**d**) segmented plastron depicted in green with annotation.

**Figure 3 animals-15-00824-f003:**
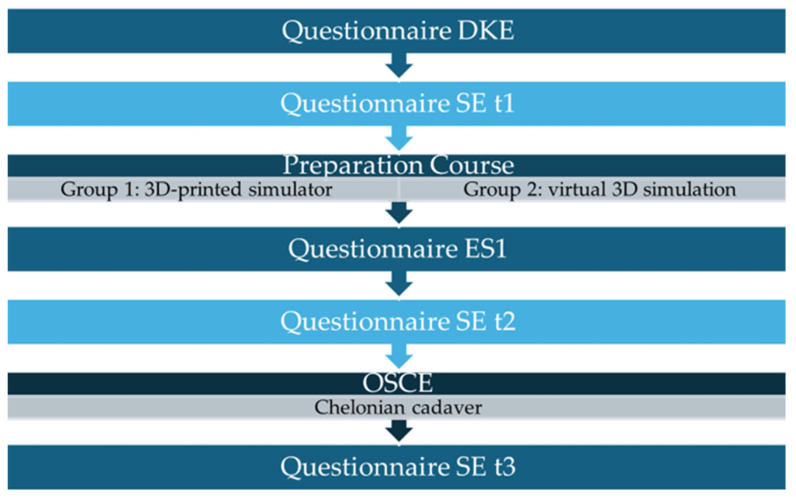
Study design depicting the process of this study for both student groups using a questionnaire on demographic data, previous knowledge, and experience (DKE), a questionnaire on self-efficacy (SE) at three points (t1, t2, t3), an evaluation sheet (ES1), and an objective structured clinical examination (OSCE) to collect the data.

**Figure 4 animals-15-00824-f004:**
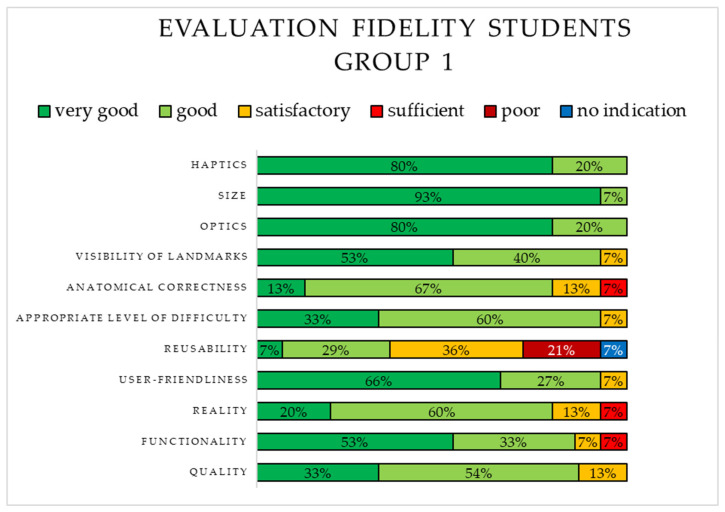
Evaluation of the fidelity of the 3D-printed simulator (S1) by students in Group 1.

**Figure 5 animals-15-00824-f005:**
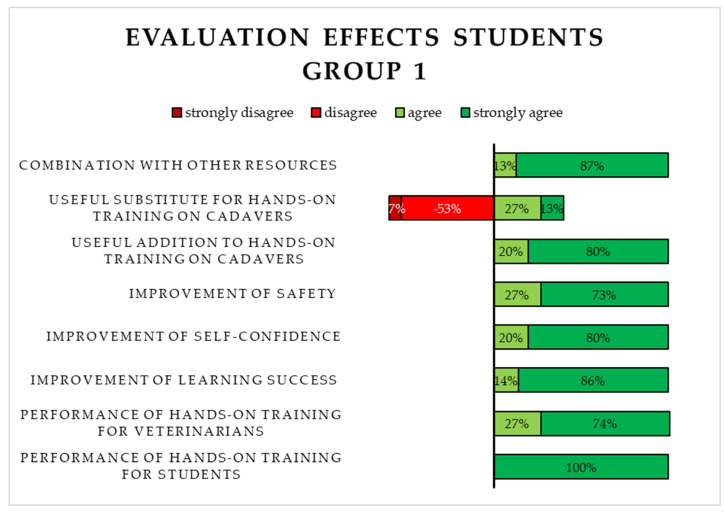
Evaluation of the 3D-printed simulator (S1) by the students in Group 1 using Likert-type items (application and effects on learning).

**Figure 6 animals-15-00824-f006:**
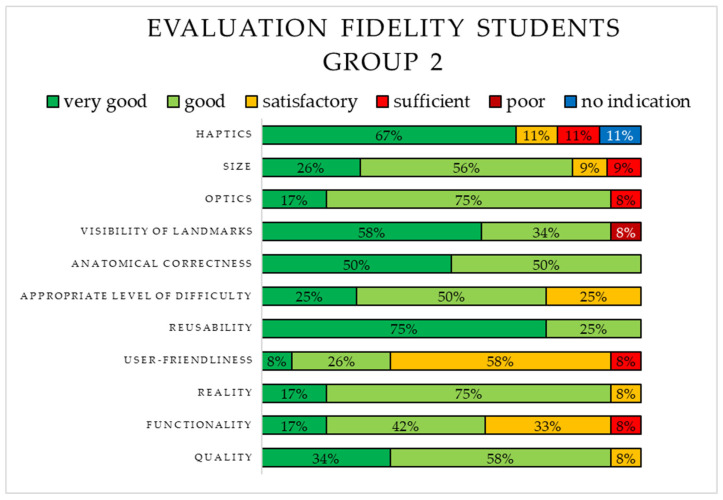
Evaluation of the fidelity of the virtual 3D simulation (S2) by the students in Group 2.

**Figure 7 animals-15-00824-f007:**
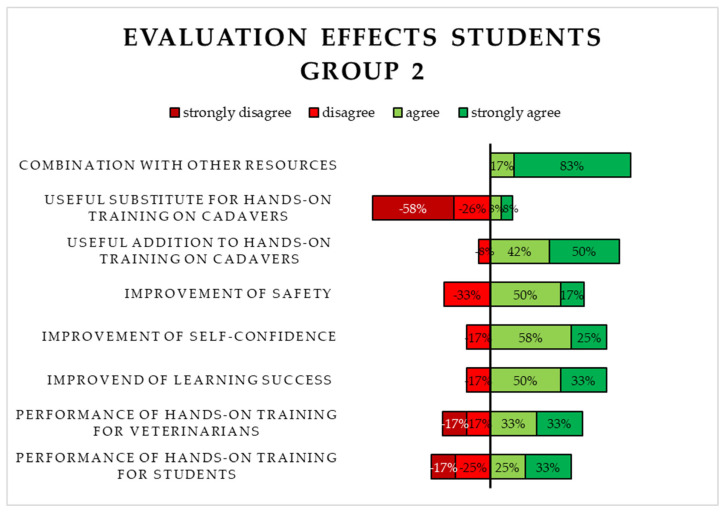
Evaluation of the 3D-printed simulator (S1) by the students in Group 2 using Likert-type items (application and effects on learning).

**Table 1 animals-15-00824-t001:** Available knowledge and experience data before the start of this study for both student groups (Group 1: 3D-printed simulator, Group 2: virtual 3D simulation).

Group	Professional Training as a Veterinary Assistant	Experience with Reptiles in Private Life	Attendance of Any Courses at the Clinical Skills Lab	Attendance of Elective “Reptile Surgery”	Theoretical Education on Reptile Handling and Diseases During Study Course	Hands-On Experience with Reptiles During Study Course
1	26%	16%	73%	0%	100%	80%
2	0%	14%	78%	0%	100%	92%
**Group**	**Coeliotomy with Veterinary Guidance**		**Coeliotomy Without Veterinary Guidance**	**Plastrotomy with Veterinary Guidance**		**Plastrotomy Without Veterinary Guidance**
1	60%		14%	16%		0%
2	78%		0%	0%		0%

**Table 2 animals-15-00824-t002:** Objective structured clinical examination (OSCE) results for both student groups (Group 1: 3D-printed simulator, Group 2: virtual 3D simulation).

Group	n	OSCE Result Mean Value	OSCE Result Standard Deviation	OSCE Result Median	OSCE Result Minimum	OSCE Result Maximum	OSCE Result Mean Score	Significance
1	15	78%	21.66	83%	45%	98%	15.7	*p* = 0.3933
2	13	72%	21.66	73%	25%	92%	13.0	*p* = 0.3933

**Table 3 animals-15-00824-t003:** Correlation of the objective structured clinical examination (OSCE) results and plastron shell thickness (ST in cm) depending on both student groups (Group 1: 3D-printed simulator, Group 2: virtual 3D simulation) including minimum (min) and maximum (max.) values.

Group	ST Max.	ST Min.	ST Median	ST Standard Deviation	OSCE Result Min.	OSCE Result Max.	OSCE Result Median	Correlation Spearman	Significance
1	0.6	0.3	0.4	0.11	45%	98%	83%	−0.31879	*p* = 0.3693
2	0.6	0.3	0.4	0.10	25%	92%	73%	−0.39789	*p* = 0.2889

**Table 4 animals-15-00824-t004:** Self-assessment of skills (SE) of item 7 (confidence in correct estimation of the depth of the cutting disk) after the preparation course and prior to (t2) the objective structured clinical examination (OSCE) as well as item 12 (confidence in performing a plastrotomy on a chelonian cadaver with veterinary supervision and guidance) and item 13 (confidence in performing a plastrotomy on a chelonian cadaver without veterinary supervision and guidance) after (t3) the OSCE for both student groups (Group 1: 3D-printed simulator, Group 2: virtual 3D simulation).

Item	Group	n	SE Mean Score	SE Standard Deviation	SE Sum of Scores	Significance
7	1	15	12.20	19.81	183.0	*p* = 0.0362
7	2	14	18		252.0	
12	1	15	12.50	13.16	187.50	*p* = 0.0250
12	2	13	16.08		218.50	
13	1	15	11.70	19.27	175.50	*p* = 0.0313
13	2	13	17.73		230.50	

**Table 5 animals-15-00824-t005:** Correlation of objective structured clinical examination (OSCE results) and self-assessment of skills (SE) after the preparation course (t2) depending on the student group (Group 1: 3D-printed simulator, Group 2: virtual 3D simulation).

Group	SE t2 Maximum (Sum of Points)	SE t2 Minimum (Sum of Points)	SE t2 Mean Value (Sum of Points)	SE t2 Median (Sum of Points)	SE t2 Standard Deviation	OSCE Result Median	Correlation Spearman	Significance
1	33	19	25	24	4.3	83%	0.14452	*p* = 0.6073
2	42	20	28	27	6.4	73%	−0.25659	*p* = 0.3974

**Table 6 animals-15-00824-t006:** Desired learning resources (S1 = 3D-printed simulator, S2 = virtual 3D simulation) depending on the evaluation group (Group 1: 3D-printed simulator, Group 2: virtual 3D simulation).

Desired Learning Resource	Group 1	Group 2	Expert Group
S1	33%	15%	60%
S2	0%	0%	0%
Combination of S1 and S2	67%	85%	40%

## Data Availability

The data presented in this study are available upon request from the corresponding author. The data are not publicly available due to data privacy.

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
