# Peer review of "Come out of Your Shell—A Comparative Pilot Study for Teaching the Central Plastrotomy in Chelonians Using a 3D-Printed Simulator and a Virtual 3D Simulation"

_animals, 2025, doi:10.3390/ani15060824_

Round 1
Reviewer 1 Report
Comments and Suggestions for Authors
This article describes a study conducted among 3rd and 4th year students to compare two ways to support training in veterinary surgery (plastrotomy) on chelonians (turtles): the use of a 3D printed physical simulator on the one hand and a 3D virtual simulation on the other hand. These two approaches constitute more or less innovative practices in teaching: these practices are already used for research and for veterinary (and human) medicine, and the use of physical anatomical atlases in medicine is quite old. One group used the simulator, and the other group used the simulation to train. Then, the students' practical skills were validated in a clinical examination and the students evaluated their respective training resource (simulator and simulation). Experienced veterinarians also evaluated (positively) the simulator. An increase in skills in both groups was observed. Students trained on the simulator performed better than those trained with simulation, but the differences between groups were generally not significant. The simulator, however, performed on average better in all test environments than the simulation The conclusion of the study is that the 3D printed simulator is a relevant educational tool for veterinary students, and is a complement to traditional methods on cadavers, therefore practices more in line with the new societal aspirations of students.
This paper is very descriptive, there are too much sharp details for hardware nomenclature, models materials, participant profiles, experiment set-ups. The results presentation and comments are much more interesting, including the student feedbacks that are clearly identified and presented. For instance, in table 8, the two student groups and the expert group show their preference for 3D-printed simulator compared to simulation alone, but they are interested in combining the two ways of training. Figure 4 and 5 are giving more details on the evaluation of the fidelity of the 3D-printed simulator by group 1 students, and the same for figure 6 and 7 respectively for group 2 students and the simulation. The conclusion, chapter 5, is too short. It is limited to an ultra-short very general sentence recalling that the findings provided by the study of the use of 3D-printed simulators and virtual 3D simulations for clinical skills learning can lead to further research. For this, content from the discussion chapter (4), for instance “limitations and strengths” part (furthermore it is the only one with a given subtitle) could be merge with the conclusion, and should be extended introducing some ideas for identified potential future works.
Some presentation issues to address:
-Internal borders in table 1
-Words badly cut in table 3
Author Response
Open Review 1
Dear reviewer, thank you for you for taking the time to read the submitted manuscript. Your comments are an important contribution to it and your feedback was very encouraging.
Comments 1: “This paper is very descriptive, there are too much sharp details for hardware nomenclature, models materials, participant profiles, experiment set-ups.“
Response 1: Thank you for your suggestion. The authors agree that the manuscript includes many details, however some passages were added, rather than shortened for better understanding. Since reviewer 4 asked for a detailed description of the procedure in the preparation course was added. Please see Line 224-231.
Comments 2: „The conclusion, chapter 5, is too short. It is limited to an ultra-short very general sentence recalling that the findings provided by the study of the use of 3D-printed simulators and virtual 3D simulations for clinical skills learning can lead to further research.
Response 2: Thank you for your suggestion. The template of animals reads concerning the conclusion “This section is mandatory, with one or two paragraphs to end the main text.” Hence to this, the conclusion will be limited to the two sentence present. However the authors tried to include your comment and added more information in the second sentence. Please see Line 748-749.
Comments 3: „For this, content from the discussion chapter (4), for instance “limitations and strengths” part (furthermore it is the only one with a given subtitle) could be merge with the conclusion, and should be extended introducing some ideas for identified potential future works“.
Response 3: Thank you for this amendment, the authors have added further sub headlines for better structure within the discussion. Please see Line 590, 649 and 734. Moreover, the further implementation of S1 was added and explained in a few sentences. Please see Line 734-743.
Comments 4: „Internal borders in table 1“
Response 4: Thank you for the comment, the authors have adjusted the format and content of all tables used in the manuscript for better legibility. Please see Table 1-8.
Comments 5 „Words badly cut in table 3“
Response 5: Please see Response 4. Additionally, abbreviations were now used in Table 3 for better legibility.
Thank you for the positive comments.
The authors hope to be able to implement the proposals made by the reviewer.
Reviewer 2 Report
Comments and Suggestions for Authors
Dear Authors,
thank you for submitting this interesting manuscript. Your study, which aims to evaluate and compare a 3D-printed simulator and a 3D virtual simulation as teaching devices for performing a central plastrotomy in chelonians, is well structured and detailed. I believe it could make an important contribution to the development of novel learning approaches for complex surgical skills such as central plastrotomy.
I just have a few minor observations:
Lines 38-39: I agree that it is a useful alternative to other teaching methods, such as the use of cadavers, but I don't see how it can affect animal welfare and species conservation. I think this consideration should be placed in the context of the animal welfare legislation of each country. In Italy, for example, the sacrifice of animals for educational purposes is not permitted. Therefore, the use of cadavers is a valid teaching method that in no way poses a risk to animal welfare or conservation. I believe that a revision of this statement could help the authors to improve the clarity of the period.
General comments on the introduction
In my opinion, the introduction is well written. However, some improvements to the structure and logical order of the topics described would make it easier for readers to understand.
Lines 49: What do the authors mean by ‘thereto’?
Lines 70-71: I think it would be better to move the aim of the study to the end of the introduction.
Lines 73-80: I think it would be better to include this period in the first part of the introductory section, where the challenges of exotic animal medicine are highlighted.
Line 113: '(Testudo graeca nabeulensis)' I think it should be in italics.
Lines 216-224: If there are 29 students, how did the authors get 10 students in each of the three semesters (30 students in total)? I think this period needs to be clarified.
Line 280: ‘collection had already been validated for previous study purposes.’ Could the authors include a bibliographical reference for these studies?
Line 376-378: see comment for lines 216-224
In the hope that my comments will provide some useful insights, thank you for your work.
Author Response
Open Review 2
Dear reviewer, thank you for you for taking the time to read the submitted manuscript. Your comments are an important contribution to it and your feedback was very encouraging.
Comments 2 „Lines 38-39: I agree that it is a useful alternative to other teaching methods, such as the use of cadavers, but I don't see how it can affect animal welfare and species conservation. I think this consideration should be placed in the context of the animal welfare legislation of each country.“
Response 1: Thank you for your suggestion, unfortunately the number of available cadavers is limited. The killing of animals for educational purposes is further limited according to German law. The authors have therefore explained more detailed why an animal-free alternative like the simulator contributes to animal welfare and species conservation by offering veterinary students more training opportunities. Please see Line 619-629.
Comments 2 „Lines 49: What do the authors mean by ‘thereto’?“
Response 2: Thank you for this amendment, the word “thereto” was deleted from the sentence. Please see Line 53-55.
Comments 3„Lines 70-71: I think it would be better to move the aim of the study to the end of the introduction.“
Response 3: Thank you for this comment, the sentences were moved to the end of the introduction. Please see Line 76-78 and 90-93.
Comments 4: “Lines 73-80: I think it would be better to include this period in the first part of the introductory section, where the challenges of exotic animal medicine are highlighted.
Response 4: Thank you for this comment, the sentences were moved to the beginning of the introduction. Please see Line 47-52.
Comments 5: “Line 113: '(Testudo graeca nabeulensis)' I think it should be in italics.”
Response 5: You are right, the Latin name is now presented in italics. Please see Line 111.
Comments 6 “Lines 216-224: If there are 29 students, how did the authors get 10 students in each of the three semesters (30 students in total)? I think this period needs to be clarified.”
Response 6: Thank you for your suggestion. The background information is, that 30 seats were allocated in the elective, but one person was absent without excuse, so that the planned allocation of 10 persons was not achieved in one of the three cycles conducted. Also one person of the 29 students did not attend the OSCE, thus reducing the number to 28. Since the section “limitation and strengths” in the discussion further explains these circumstances in detail (“The fact that one person did not take part in the elective course and another student did not participate in the OSCE was another limitation concerning the sample size“), the authors decided to not additionally mentioning it in the prior sections, which is in line with the suggestion of reviewer 1 that the manuscript is already too detailed.
Comments 7: “Line 280: ‘collection had already been validated for previous study purposes.’ Could the authors include a bibliographical reference for these studies?”
Response 7: Yes, thank you for the amendment, the references have been added and the sentence was adjusted. Please see Line 284-286 and 311-312.
Comments 8 “Line 376-378: see comment for lines 216-224”
Response 8: Please see Response 6.
Thank you for the positive comments.
The authors hope to be able to implement the proposals made by the reviewer.
Reviewer 3 Report
Comments and Suggestions for Authors
A manuscript (animals-3500098) entitled “Come out of your shell - A comparative pilot study for teaching the central plastrotomy in chelonians using a 3D-printed simulator and a virtual 3D simulation” authored by Marie-Therese Knoll, analyzes the use of a 3D-printed simulator and a virtual 3D simulation for teaching the central plastrotomy in chelonians (turtles and tortoises), comparing the effectiveness of these methods as educational tools for veterinary students. The items described in the manuscript meet the scope of the journal Animals, under “animal science” and “method of study”. Sixty-four references are used to construct the article, of which the oldest is from 2005. Most are quite current, however the authors use many company websites, conferences and even YouTube videos. The articles cited were published in important journals. There are two papers in which two co-authors of the study are also co-authors, but without a conflict of interest in the production of the material in question. The introduction raises the issue of teaching techniques to exotic animals, which are increasingly present in homes as unconventional pets, justifying the study. However, a curious thing is that the authors place the objective of the study in the introduction in an unconventional place (lines 70 and 71), before the full basis of the study (lines 87 and 88). This may be a new trend in writing articles, but it may seem strange to the reader. One suggestion is to try to adjust the final third of the introduction, placing the objective of the study closer to the end of the text, without compromising the explanatory logistics. This is not a request from this reviewer, but a suggestion, as it may be a descriptive innovation, but more traditional readers may find this writing style strange. The methodology is quite detailed, with regard to the procedures developed and executed, with sources provided in an adequate and complete manner. The statistical tests are adequate and their use is well explained. Table 1 needs to be better designed, but I believe this is a problem in the editing of the document, which is the responsibility of the journal's designers and not the authors. However, both should check the cause of the problem and fix it. Table 2 highlights the p value, with 4 decimal places, demonstrating adequate precision. Table 3 requires attention to the text's aesthetics, with letters of the same word being separated inappropriately. This is merely an aesthetic issue. Better adjustment of the column widths would be very welcome. The line spacing could be better adjusted in all tables. Figure 4 could be improved aesthetically. The finishing lines of the letters and the graph are rough. The size could be better adjusted, centered. If possible, refine the aesthetics. The same happens in figures 5, 6 and 7. One fact already observed in the discussion is the different spacing formatting between the discussion and the other items. Checking the template provided by the journal, there is no spacing, neither before nor after the paragraphs. The line spacing is 14 points, and is well-allocated, legible, and understandable. The discussion is very detailed and justifies the weak points of the study, especially, in my opinion, because the sample was small, which could have influenced the statistical analysis. Perhaps a larger N would have made a significant difference. In conclusion, the study is innovative and meets an important demand regarding the use of animals in practical classes, seeking to replace them with alternative methods, without compromising teaching and learning. In addition, it is a method that can be applied in many places, especially with the dissemination of 3D printing technology. As negative points, the study did not demonstrate any difference between the factors studied, but the N was small. Also in this context, the dependence on technology, despite being a factor that cannot be avoided in the future, is still a limiting factor for many, especially in underdeveloped countries. However, the study opens doors for new research and references technologies that are being implemented in new teaching and learning models in veterinary medicine. Another strong point was the revision in English, which made it pleasant to read and easy to understand. References should be checked again for formatting errors.
To the authors:
• On line 102, insert a comma between the terms “Germany” and “before”.
• On line 113, the scientific name should be in italics.
• On line 116, insert a comma between the terms “China)” and “and”.
• On line 136, insert a comma between the terms “Germany)” and “was”.
• On line 141, insert a comma between the terms “Spain)” and “were”.
• On line 887, the journal presented as a reference was incorrectly identified as Pesq Vet. However, it should be Pesq Vet Bras.
Author Response
Open Review 3
Dear reviewer, thank you for you for taking the time to read the submitted manuscript. Your comments are an important contribution to it and your feedback was very encouraging.
Comments 1: “Table 1 needs to be better designed, but I believe this is a problem in the editing of the document, which is the responsibility of the journal's designers and not the authors. However, both should check the cause of the problem and fix it.”
Response 1: Thank you for the comment, the authors have adjusted the format and content of all tables used in the manuscript for better legibility. Please see Table 1-8.
Comments 2: “Table 3 requires attention to the text's aesthetics, with letters of the same word being separated inappropriately. This is merely an aesthetic issue. Better adjustment of the column widths would be very welcome. The line spacing could be better adjusted in all tables.”
Response 2: Please see Response 1.
Comments 3: “Figure 4 could be improved aesthetically. The finishing lines of the letters and the graph are rough. The size could be better adjusted, centered. If possible, refine the aesthetics. The same happens in figures 5, 6 and 7.”
Response 3: Thank you for your suggestion, all figures were centered. Please see Figure 4-7. Further refinements in the aesthetics were unfortunately not possible.
Comments 4: “One fact already observed in the discussion is the different spacing formatting between the discussion and the other items. Checking the template provided by the journal, there is no spacing, neither before nor after the paragraphs.”
Response 4: Thank you for your amendment, the formatting was adjusted in the whole manuscript to 14Pt. without spacings before and after the paragraphs.
Comments 5: “References should be checked again for formatting errors.”
Response 5: Thank you for your comment, the formatting was checked and it was identified that the uneven spacing between some of the references was due to the justification.
Comments 6: “On line 102, insert a comma between the terms “Germany” and “before”.”
Response 6: Thank you for your amendment, the comma was inserted. Please see Line 100.
Comments 7: “On line 113, the scientific name should be in italics.”
Response 7: Yes, you are correct, the Latin name is now presented in italics. Please see Line 111.
Comments 8: “On line 116, insert a comma between the terms “China)” and “and”.”
Response 8: Thank you for your amendment, the comma was inserted. Please see Line 114.
Comments 9: “On line 136, insert a comma between the terms “Germany)” and “was”.”
Response 9: Thank you for your amendment, the comma was inserted. Please see Line 133.
Comments 10: “On line 141, insert a comma between the terms “Spain)” and “were”.”
Response 10: Thank you for your amendment, the comma was inserted. Please see Line 138.
Comments 11: “On line 887, the journal presented as a reference was incorrectly identified as Pesq Vet. However, it should be Pesq Vet Bras.”
Response 11: Thank you for your suggestion, the reference was updated. Please see Line 930.
Thank you for the positive comments.
The authors hope to be able to implement the proposals made by the reviewer.
Reviewer 4 Report
Comments and Suggestions for Authors
This paper is well written and easy to read. The rationale and novelty of the study are well established. The following suggestions are offered for its improvement:
Abstract: Following the journal’s instructions for authors, please add 1-2 sentences on the study’s Background: Place the question addressed in a broad context and highlight the purpose of the study.
Method: L227 - Both conditions were described in detail regarding their technical specifications. However, it is not clear what the students did in both cases. Please explain in more detail how the hands-on training was organized.
L284: Regarding the OSCE, is the protocol followed novel, or has it been used before? Please mention here relevant works.
The results analysis is appropriate.
Author Response
Open Review 4
Dear reviewer, thank you for you for taking the time to read the submitted manuscript. Your comments are an important contribution to it and your feedback was very encouraging.
Comments 1: “Abstract: Following the journal’s instructions for authors, please add 1-2 sentences on the study’s Background: Place the question addressed in a broad context and highlight the purpose of the study.”
Response 1: Thank you for you suggestion. The abstract was partly rewritten to explain the aim of the study while not exceed the number of 200 words. Please see Line 27-29, 34-35 and 37-39.
Comments 2: “Method: L227 - Both conditions were described in detail regarding their technical specifications. However, it is not clear what the students did in both cases. Please explain in more detail how the hands-on training was organized.”
Response 2: Thank you for your comment, a more detailed description of the procedure in the preparation course was added. Please see Line 224-231.
Comments 3: “L284: Regarding the OSCE, is the protocol followed novel, or has it been used before? Please mention here relevant works.”
Response 3: Thank you for the amendment, the references have been added and the sentence was adjusted. Please see Line 291-293
Thank you for the positive comments.
The authors hope to be able to implement the proposals made by the reviewer.